# Driving innovation: When design collaboration becomes open source, how do reward mechanisms and in-process feedback act as catalysts

Boqun Xu [1,2] , Kexin Xu [1] , Yuning Zhu [1]*, Yiting Zhou[1‡], Xiaojian Liu[3]

**1** Department of Design and Architecture, Zhejiang University of Technology, Hangzhou, China,
**2** Zhejiang Dafeng Industry Co., Ltd., Hangzhou, China, **3** Industrial Design and Research Institute, Zhejiang University of Technology, Hangzhou, China

☯ These authors contributed equally to this work.
‡ TZ and XL authors also contributed equally to this work.
* zhuyuning@zjut.edu.cn

## Abstract

Open innovation, prevalent in projects like design crowdsourcing, is widely acknowledged for its role in advancing technology through knowledge flow. However, the competitive nature of design teams often keeps core knowledge confidential, hindering collaboration in the field. Open-source ideation in software facilitates resource sharing, which has been demonstrated to overcome information barriers, fostering intellectual progress and driving remarkable outcomes. If applied to product design, could it make collaborative innovation more efficient or more difficult? External incentives are key in this dynamic. This study focuses on two representative motivational factors: reward mechanisms and transparent in-process feedback, and conducts a 2 × 2 experiment where designers completed multiple rounds of product colorization tasks with simultaneous resource exchange in an online environment. We evaluate design outcomes, collaborative referencing networks, and individual behaviors as key characteristics. Results indicate that the Quota Allocation Method (QAM) increases individual productivity (quantity) and competitive awareness, while the Contribution-Based Allocation Method (CAM) fosters creativity (quality) and long-term engagement, along with higher referencing density and satisfaction. Offering in-process feedback further optimizes collaboration outcomes but may introduce profit-driven behaviors in the CAM setting. By revealing the interaction between incentive mechanisms and individual motivations in open-source design collaboration, this study offers practical strategies to advance previous theories, promoting the effective implementation and sustainable collaboration of open-source models in product design.

**Data availability statement:** All relevant data are within the manuscript and its Supporting Information files.

**Funding:** Xu B.: Zhejiang Natural Science Foundation (Grant No. LQ22F020029). URL:https://zjnsf.kjt.zj.gov.cn/portal/index.html; Philosophy and Social Science Planning Project of Zhejiang Provincial (Grants No. 23NDJC091YB). URL: https://www.zjskw.gov.cn/ Zhu Y.: Philosophy and Social Science Planning Project of Zhejiang Provincial (Grants No. 24NDJC151YB). URL: https://www.zjskw.gov.cn/ The funders had no role in study design, data collection and analysis, decision to publish, or preparation of the manuscript.

**Competing interests:** The authors have declared that no competing interests exist.

## Introduction

Open innovation is a paradigm that uses purposive inflows and outflows of knowledge to accelerate innovation [1], offering advantages like low cost, borderless collaboration, and efficient resource aggregation [2,3]. This approach manifests as crowd creativity in product design, with crowdsourcing and competitions being the popular modalities. They typically outsource the missions to a large network of potential laborers through an open call, which usually takes the form of peer-production (when the job is performed collaboratively), but is also often undertaken by sole individuals [4]. In recent years, the collaborative ideation associated with open source software (OSS) has introduced new possibilities for crowdsourcing, positing that in collaboration, knowledge sharing allows participants to draw inspiration from others' ideas [5,6], often leading to breakthrough contributions [7]. However, it can also result in management difficulties and cause confusion [8,9]. Open-source collaboration shares similarities with open innovation, characterized by diverse participants and a transparent process. Numerous researchers have explored and affirmed the effectiveness of combining open source thinking with innovative design, such as using collaborative genetic algorithms, demonstrating that open-source collaboration can optimize design outcomes and stimulate participants' creative potential [10,11]. Nevertheless, in real-world projects, competition for interests often leads to unpredictable collaborative outcomes. According to motivation theory [12], an organism's perception of external rewards, constraints, the nature of feedback and other environmental factors can significantly influence intrinsic motivation [13], which, in turn, may affect project effectiveness.

Among the various conditions impacting innovation and collaboration, reward mechanisms are believed to effectively activate participants' motivation [14–18]. However, participants often worry whether the benefits they receive are reasonably proportional to their efforts [19], and due to the difficulty in balancing knowledge sharing and knowledge protection [20], their behavior and attitudes may be disrupted, impacting the creativity of the outcomes. Moreover, studies in various fields have demonstrated that providing in-process feedback can enhance project performance [21–23]. By offering transparent directional guidance, it reduces participants' perceived uncertainty and consequently enhances creativity and engagement [24–26]. However, the effects are multifarious; in competitive settings, in-process feedback as a real-time leaderboard may suppress participants' competitive spirit and drive [27–30]. Current literature has not yet achieved consensus regarding the predominant factors that intervene in or dominate the open innovation process [31].

This study focuses on the open-source collaboration model in product design, exploring the application and value of "drawing inspiration from shared creation". By simulating open-solution design tasks that allow mutual referencing, the research examines the impact of reward mechanisms and process feedback as boundary conditions on innovation performance, such as the quantity of design works and their intergenerational evolution characteristics. Additionally, by observing differences in participants' behavior and attitudes, the study analyzes strategy adjustments and decision-making processes triggered by changes in rules to elucidate the evolution

and impact of individual and collective creativity. This research provides new insights on how to optimize collaboration rules, maximize project efficiency, and improve participant experience, making it an important attempt to guide the effective implementation of open-source design collaboration in product crowdsourcing activities.

## Literature review

### Open-source design collaboration

Traditional creative activities were usually conducted by individuals or small groups. In contrast, contemporary open innovation, such as crowdsourcing [32] and competitions, harnesses broader collective collaboration to address challenges, make decisions, and generate new knowledge, and have been extensively studied across various fields [33–35]. Among these approaches, Open Source Collaboration (OSC) stands out by fostering sustained community engagement and facilitating idea exchange through knowledge sharing and voluntary contributions, emerging as an impactful contemporary model for innovation. Raymond (1999) argued that open source intelligence would ultimately produce better outcomes than proprietary software. The success of platforms like GitHub and Linux has validated this perspective [36,37]. Although open source may present security issues such as intellectual property and personal privacy concerns [9], as well as challenges like inefficiency, disorder, and maintenance difficulties [8], it remains a powerful force driving the co-evolution and development of both products and communities, often leading to breakthrough advancements [38,39].

The origins of open-source practices in design can be traced to brainstorming [40], which combines multiple ideas to generate new ones. Shared ideas were believed to foster collaborative thinking and produce results superior to individual efforts [41–43]. Members benefit from each other's insights [44] and produce more meaningful creations [45], with the value of group imitation also being recognized [46]. Crowdsourcing has been widely applied in design activities, such as Dell's IdeaStorm community, which has generated numerous valuable ideas through continuous contributions. This creative process aligns with the open source mindset, emphasizing openness, collaboration, and community-driven characteristics. Researchers have explored open-source design crowdsourcing, developing methods such as the collaborative approach that integrates crowd sketching processes [47] and the crowdsourced idea generation model [48]. Some scholars have also applied group-based open-source collaboration to product design, demonstrating its positive impact on project outcomes and participants' innovative potential [10,11]. Open-source transparency and democratization can be effectively integrated into viable models through crowdsroucing, fostering innovation and profitability [14]. When integrated with product development, it demands careful planning, particularly in areas such as task allocation and organizational structure [49, 50]. Considering task authenticity and interest-based rules, open-source mechanisms and group behaviors in product design collaboration warrant further exploration.

### Reward mechanisms

Rules are crucial for maintaining human cooperative relationships. To ensure the stability and success of collaboration, it is essential to establish appropriate rules that balance the relationships between participants and the project [51,52].

Contributions to open-source collaboration are typically considered voluntary, primarily driven by intrinsic motivation and the intention for social benefits, such as satisfaction, achievement, and hedonic motivation [53,54]. As research has progressed, reward mechanisms have been recognized as crucial for enhancing participant enthusiasm and the quality of collaborative outcomes [17,18]. Rewards can be monetary, such as cash, or non-monetary, like recognition, positions, etc. [55,56]. In crowdsourcing competitions, monetary prizes serve as a universal material incentive for human capital, effectively activating extrinsic motivation [14,16], attracting more participants [57] and increasing the quantity of contributions [58]. The positive impact of monetary rewards has been proven in open source communities [15], but the amount also needs to be considered. Higher rewards may intensify competition, potentially compromising project quality and reducing individual motivation to contribute [59].

The effectiveness of rewards is grounded in the fairness expectation theory, which states that people believe the distribution of project benefits is fair [60,61]. Participants often express concerns about whether the benefits they receive are reasonably proportional to their efforts [19]. In an open-source environment, balancing knowledge sharing with knowledge protection presents a significant challenge [15]. Therefore, reward distribution strategies must be properly arranged. A study in crowdsourcing competitions found that allowing participants to choose personalized rewards motivated them to exert more effort and produce better solutions [17]. Additionally, a blockchain-based incentive mechanism for open-source group contributions has been proposed, which measures the value of developers' contributions and incentivizes continued participation in communities through a virtual points system [62], shifting from equal distribution to a performance-based model. While rewards can promote collaboration, they may also have the potential to suppress effort. Specific compensation models can be considered for different teams [63].

## In-process feedback

Collaboration is a cyclical process where ideas are continuously evaluated and revised, necessitating the timely provision of in-process feedback. Such environmental transparency not only guides the project towards specific needs and better outcomes but is also a key factor in influencing participant motivation [22,25,64].

Feedback comes in various forms, with evaluation being a typical form. The quantity and nature of evaluation can serve as feedback indicators [65], or real-time performance ratings can be implemented [66,67]. For instance, design crowdsourcing platforms such as 99designs and Crowdspring enable project owners to assign ratings to submissions using a five-star system. Ideas generation and exchange are critical to the innovation process. Evidence suggests that low-evaluation environments can enhance creativity within groups [25,68]. Moreover, in-process feedback plays a distinctive role in competitive contexts by revealing participants' standings and the competitors' levels, similar to a "leaderboard," which is one of the simplest mechanisms for fostering interaction [29,69]. Some studies suggest that leaderboards can stimulate competitive spirit, enhancing motivation and performance [27,30], while in other cases, they may be ineffective or even have negative effects [28,68]. An online competition experiment reported that directed feedback benefits the average quality of entries submitted, while no feedback or random feedback may produce better top-end entry quality [67].

An individual's intentions and behaviors are influenced by the utility of feedback [70]. It is worth noting that the perceived utility of feedback can affect motivation toward its acceptance and usefulness [71]. For example, recipients may accept in-process feedback to enhance performance and meet project expectations, but if they find the feedback unhelpful or unsatisfactory, they may ignore it [72,73]. The academic community has examined the impact of feedback on participant behavior across various related contexts, demonstrating that in-process feedback can guide participants by revealing project owners' preferences and improving proposal quality. However, it may also expose performance gaps, which could affect motivation and engagement in later stages [67].

## Summary

Numerous factors influencing the creative outcomes and participant attitudes in crowdsourcing projects have been widely discussed, including extrinsic motivations like material or non-material rewards [16,74], as well as intrinsic motivations like a sense of achievement [75–77]. Timely in-process feedback also holds significance. The high transparency and complex competitive dynamics of open-source collaboration necessitate the establishment of clear rules to ensure optimal innovation outcomes, enhance project performance, and guide team behavior. Previous research on open innovation tended to ignore the importance of business models [78]. This deepens our consideration of the current practices in open-source design collaboration, prompting this study to introduce external constraints to move beyond idealized scenarios.

The study investigates how reward mechanisms and in-process feedback influence creative outcomes and collaborative behaviors when product design supports open-source collaboration. Specifically, we want to investigate the following questions:

   

1.  How different reward mechanisms impact design output, and through what forms of individual motivation this influence occurs;

2.  Whether in-process feedback acts as a form of informational guidance that enhances or inhibits innovation and collaboration efficiency;

3.  How reward mechanisms and in-process feedback influence the structure of creative iteration networks.

An empirical experimental approach was employed for this exploratory study. The experiment adopted a 2 × 2 factorial design, designers were recruited to participate in online incentivized creative projects, where participants were divided into two groups with distinct reward allocation methods and implementing two tasks to introduce feedback as an independent variable. designers were recruited to participate in online incentivized creative projects, where participants were divided into two groups with distinct reward allocation methods and implementing two tasks to introduce feedback as an independent variable.

## Methods

### Participants

The recruitment period for this study started on November 6, 2023, and concluded on December 24, 2023. The sample was selected from relevant majors at a Chinese university and a design company, comprising 20 participants (11 male, 9 female) with an average age of 25 years. All participants had completed the necessary theoretical courses and were capable of independently executing design tasks, the personnel characteristics are shown in Table 1. They were divided into two groups, each consisting of 8 students and 2 professional designers, with efforts made to balance gender and technical skills. This combination reflects a realistic crowdsourcing project involving designers of varying expertise levels. The experiment was carried out anonymously, with participants identified only by unique codes (Group 1: A-J, Group 2: a-j). Ethical approval for this study was obtained as a written document from the School of Design and Architecture, Zhejiang University of Technology (reference no.11372/2023). Oral informed consent was recorded by audio recording from all participants before the study due to its online nature. It covered participation, anonymized data use, and result publication. Their identities were kept unknown to one another, and they were not allowed to discuss the experiment with others.

### Measures

**Experimental task.** Colorizing product is a common design mission. The experiment included two colorizing tasks for a round fan and a folding fan (Fig 1), based on standardized files and conducted using CorelDRAW software. Each task consisted of 5 rounds, with one round executed per day, and the two tasks were spaced two weeks apart to minimize fatigue effects. The color schemes from each round were publicly displayed and allowed to be referenced under certain rules, providing an open-source way. The task rules were as followed.

**Table 1. Participants' descriptive characteristics.**

| Group | Type | Personnel characteristics | | | | | | | | | |
|---|---|---|---|---|---|---|---|---|---|---|---|
| Group 1 (QAM) | Code | A | B | C | D | E | F | G | H | I | J |
| | Gender | M | F | M | M | M | M | F | F | F | F |
| | Years of Engagement in Design | 18 | 9 | 3 | 7 | 7 | 7 | 3 | 4 | 7 | 6 |
| Group 2 (CAM) | Code | a | b | c | d | e | f | g | h | i | j |
| | Gender | M | M | M | M | M | M | F | F | F | F |
| | Years of Engagement in Design | 19 | 6 | 3 | 7 | 7 | 7 | 3 | 4 | 7 | 6 |

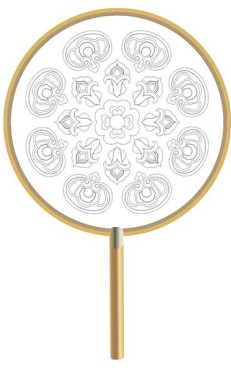
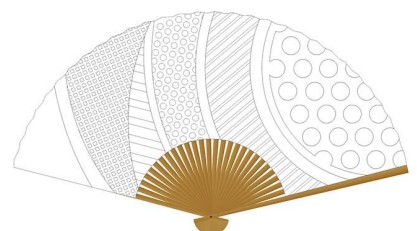

Task 1: Round fan          Task 2: Folding fan

**Fig 1. Colorizing objects.**

(1) The fan's surface contains 6 groups that can be filled with 2–6 solid RGB colors, ensuring uniform color within each group.

(2) Referencing refers to drawing inspiration from other designs, such as specific colors or techniques. Referencing is optional; however, if you choose to reference a design, please label it with its ID.

**Variables and rules.** In reward allocation, two important mechanisms were employed: **the Quota Allocation Method (QAM)**, which typically sets a fixed number of participants and corresponding prizes, and **the Contribution-Based Allocation Method (CAM)**, which distributes prizes not only to the winning entries but also based on the contributions of other submissions. We followed previous practices by using references as the basis for determining contributions (37), and established the allocation rules (see Table 2). In both tasks, the top three design works with the highest scores in each group were awarded. Group 1 used QAM, while Group 2 used CAM. Each group had a total prize pool of 500 RMB, and participants who won multiple awards had their prizes accumulated. Additionally, all participants received a USB flash drive as encouragement to prevent negative attitudes [79].

The scores of color schemes were established as feedback in this experiment, which were provided to participants in Task 2. Research indicates strong internal consistency in expert evaluations of creativity [80,81]. Drawing on the Consensus Assessment Technique [82,83] and the evaluation criteria for product color imagery [84], two senior design experts were invited to form a color scheme evaluation panel and collaboratively developed the scoring criteria (Table 3). The aesthetic score reflects one's color-matching skills and design sensibility, which develop over the long term. The technical

**Table 2. Prize allocation rules.**

| Reward mechanisms | Prize amounts and allocation rules |
|---|---|
| Quota allocation | The top-scoring work will receive **300 RMB**.<br>The second-scoring work will receive **150 RMB**.<br>The third-scoring work will receive **50 RMB**. |
| Contribution-based allocation | The top-scoring work will receive **125 RMB**, with another **125 RMB** divided equally among the designers of its reference chain.<br>The second-highest scoring work will receive **75 RMB**, with another **75 RMB** divided equally among the designers of its reference chain.<br>The third-highest scoring work will receive **50 RMB**, with another **50 RMB** divided equally among the designers of its reference chain. |

score represents the difficulty of colorizing, showcasing a person's innovation and openness to novelty, which can improve quickly through short-term learning.

## General procedure

The study utilized a 2×2 mixed experimental design, with the reward mechanism (QAM or CAM) as a between-subjects factor and in-process feedback (whether or not scores were provided) as a within-subjects factor. The experiment was conducted online according to the process depicted in Fig 2. Group chats were established on the DingTalk platform for submission collection and information feedback. Before starting, the administrator distributed tasks, materials, and introduced the rules to each group. After each round, the design works were compiled and submitted to the evaluation panel for scoring, and were provided to the corresponding group before the next round (with score feedback included in Task 2). The winning results were announced after all five rounds. A final qualitative analysis was conducted to identify the compensatory strategies participants used.

## Results

### Design outcomes

All color schemes obtained from the experiment are listed in Fig 3, corresponding to the group, task, and the number of schemes in each round.

**The quantity of design works.** Fig 4 shows that Group 1 consistently produced more schemes than Group 2 across both tasks, suggesting that QAM stimulates higher output. After score feedback was introduced in Task 2, the total number of design works decreased in both groups, with a more significant reduction in Group 2, indicating that score feedback is not conducive to generating a higher quantity of designs and has a greater impact on CAM. The standard deviations (σ) of the average number of works per person per round reveal that Group 2 exhibited greater output stability compared to Group 1, and both groups showed increased stability after feedback was provided (Task 1: $\sigma_1 = 1.3$, $\sigma_2 = 0.89$; Task 2: $\sigma_1 = 0.84$, $\sigma_2 = 0.46$). A detailed

**Table 3. Scoring criteria for color schemes.**

| Scoring metrics | Rules |
|---|---|
| Aesthetic score | **Subjective assessment (6 points)**: Experts use their professional judgment to evaluate the design's aesthetic qualities and visual harmony. |
| Technical score | **Hue technique (2 points)**: Using three or more distinct colors earns 2 points; using complementary or contrasting colors earns 1 point; using achromatic, monochromatic, analogous, or adjacent colors earns no points.<br>**Value technique (1 point)**: Adjusting the brightness of the hues earns 1 point; directly selecting colors earns no points.<br>**Saturation technique (1 point)**: Adjusting the saturation of the hues earns 1 point; directly selecting colors earns no points. |

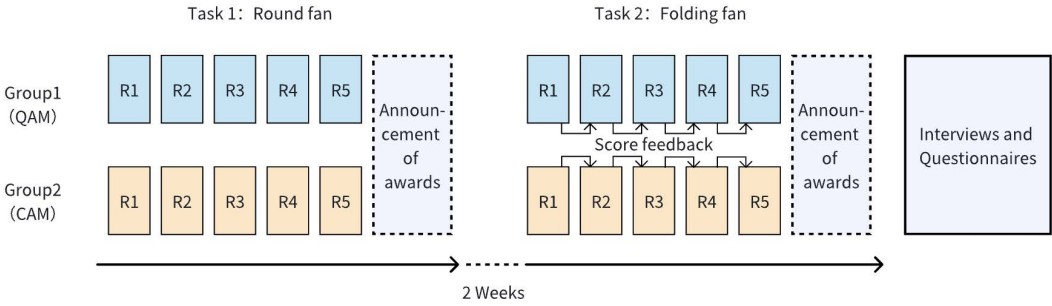

**Fig 2. Experimental procedure.**

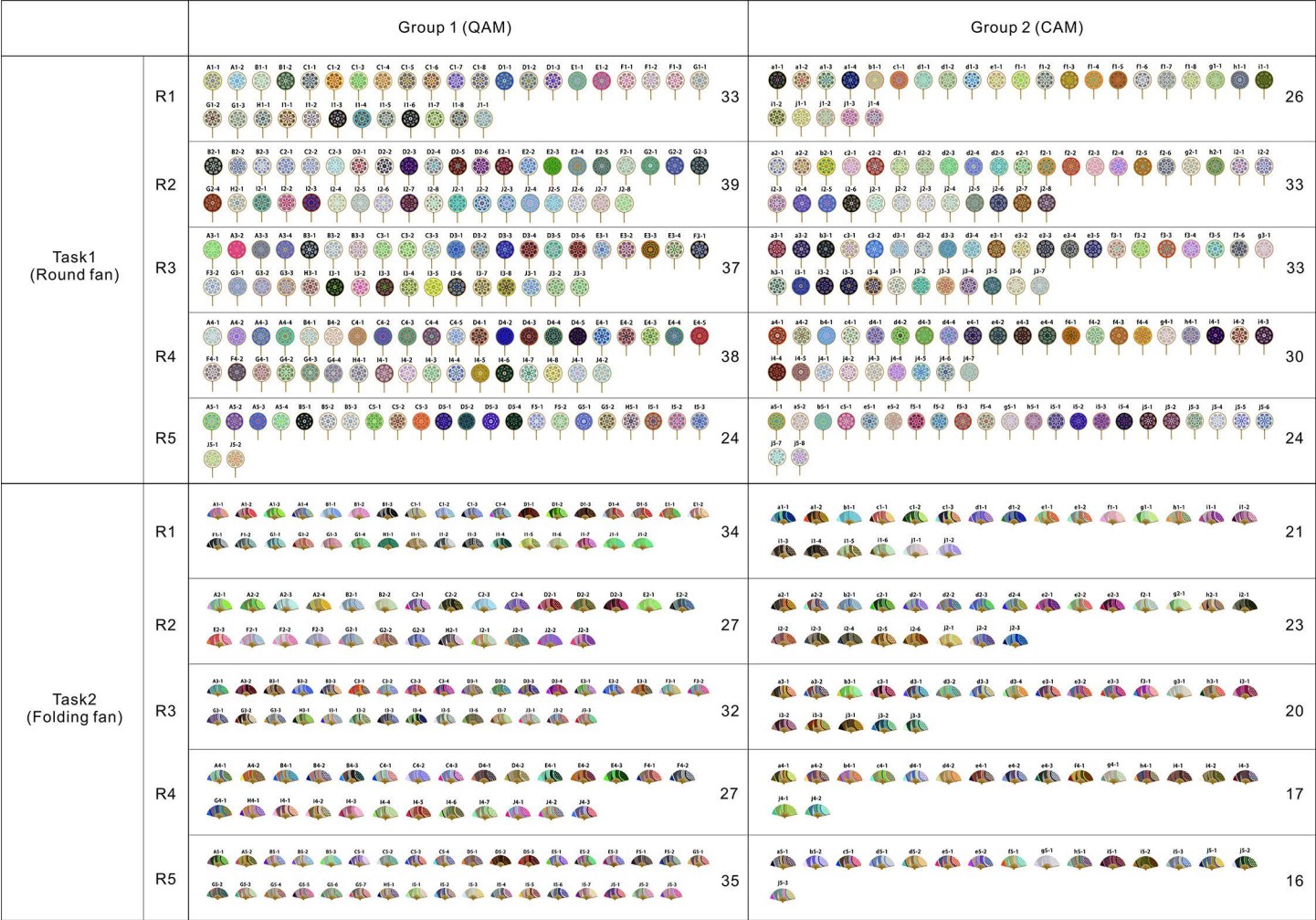

**Fig 3. Color schemes produced by each group in the two tasks.**

analysis in Fig 5 shows that participants f and j in Group 2 significantly reduced their output (f: 28–5, j: 34–13), leading to a decrease in overall group output. The reasons for these outliers will be further explored in follow-up interviews.

**The quality of design works.** The quality of design works is a key objective of open-source collaboration. Fig 6-a displays the average scores of schemes in each round and shows that Group 2 consistently outperformed Group 1 in the same task (except for round 5 in Task 2). The average scores of all design works were also higher in Group 2 (Task 1: AVG2 = 14.23, AVG1 = 12.77; Task 2: AVG2 = 12.66, AVG1 = 11.97). Given that participants had similar design abilities, it can be concluded that CAM stimulates the production of higher-quality works.

Additionally, the average scores of works in both groups significantly decreased in Task 2 and were lower than those in Task 1. Does in-process score feedback inhibit the creation of high-scoring designs? Analysis of the average score trends shows that in Task 1, where no feedback was provided, the total scores of both groups exhibited stable fluctuations. Group 1 maintained lower overall scores with a slight upward trend, while Group 2 had higher overall scores but demonstrated a slight downward trend. In contrast, Task 2 showed that the average scores of both groups exhibited slight fluctuations but followed a noticeable upward trajectory. This suggests that providing in-process feedback facilitated iterative

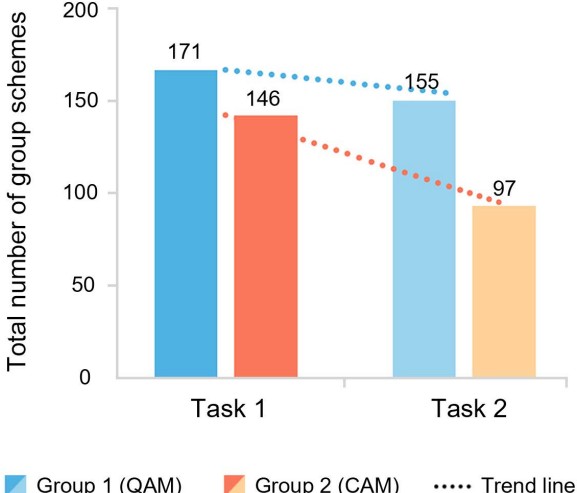

Fig 4. **The number of group schemes in the two tasks.**

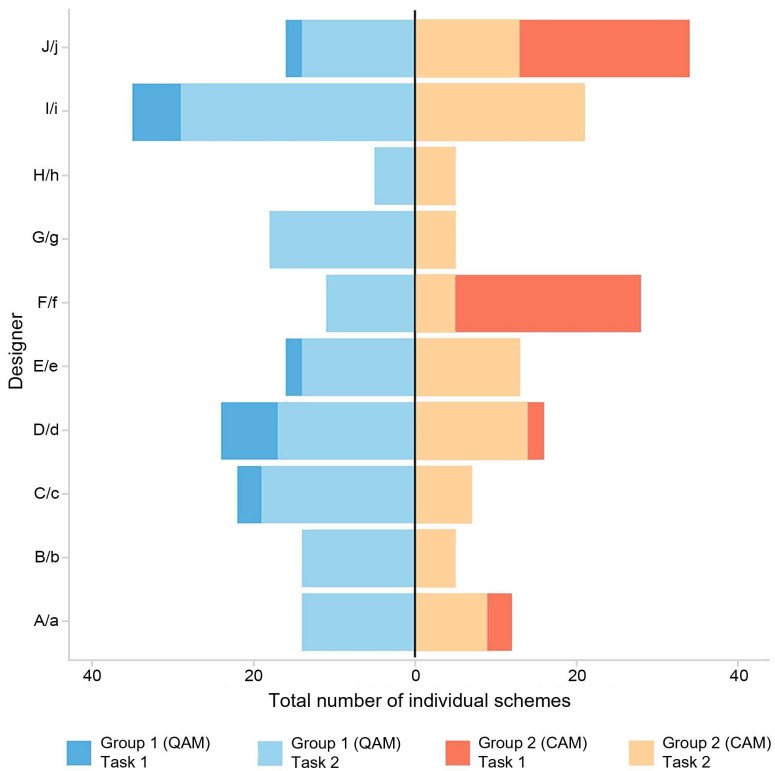

Fig 5. **The number of individual schemes in the two tasks.**

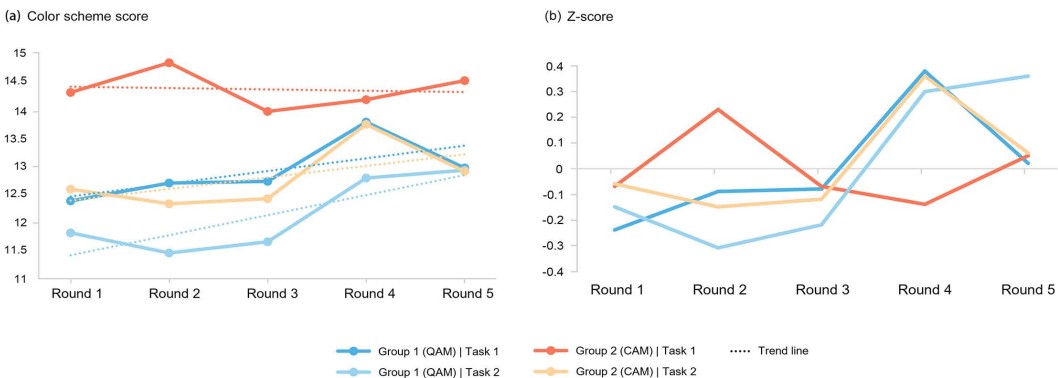

**Fig 6. Average score of group schemes in each round (a) and the standardized average score (b).**

optimization and enhancement of design quality. The trendline slopes further indicate that this feedback had a stronger motivational effect on participants under QAM (K1 = 0.36, K2 = 0.21). As the rounds progressed, the average scores of both groups converged by round 5.

To eliminate the scale effect in the original scores, the data were standardized (Fig 6-b). The results present that the quality of outcomes in Task 1 exhibited significant fluctuations, whereas in Task 2, both groups showed an upward trend, with Group 1 demonstrating notable improvement. Group 1 consistently enhanced its performance from round 2 onward and achieved optimal results by round 5, further reaffirming the positive relevance of providing in-process score feedback on quality.

Upon analyzing the two scoring metrics (see Fig 7), a significant difference was observed in the aesthetic scores between the two tasks. This may explain why the overall quality in Task 2 was lower than in Task 1, possibly due to the shift in design objectives. The presence or absence of feedback had little impact on aesthetics, as this skill requires sustained cultivation. In contrast, the technical scores showed that Group 2 consistently outperformed Group 1, indicating that CAM had a stronger effect on promoting innovative challenges and formal breakthroughs. Additionally, in Task 2, the technical scores in both groups shifted from the fluctuations seen in Task 1 to a more consistent upward trend (except for a slight decrease in Group 2 in round 5). This clear improvement suggests that providing in-process score feedback helped participants identify high-quality designs, encouraging them to quickly focus and enhance their creative difficulty.

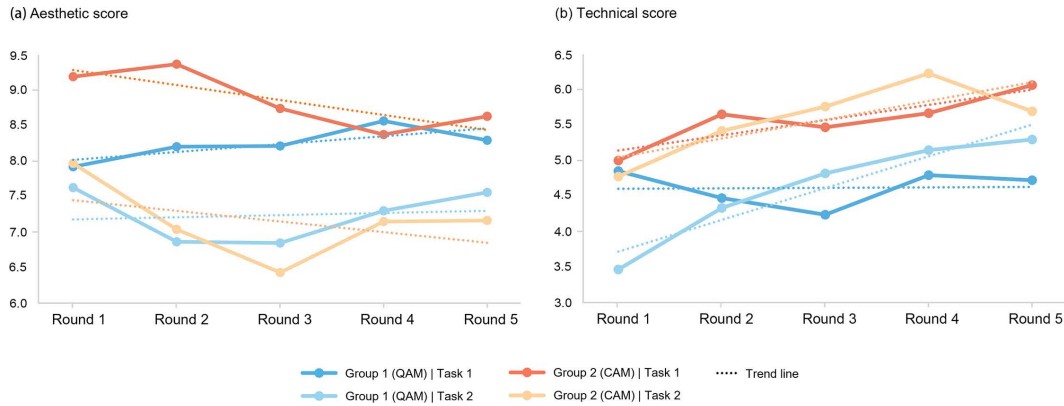

**Fig 7. The average aesthetic score (a) and technical score (b) of group schemes in each round.**

**Summary.** The results suggest that QAM incentivized participants to produce higher output, though the quality was less remarkable. This may be due to the limited number of three fixed beneficiaries in this mechanism, prompting participants to adopt a quantity-over-quality strategy to maximize their chances of winning. In contrast, CAM encouraged the production of higher-quality works, albeit with a reduced number of submissions. This reduction could be attributed to the perception that rewards are easier to obtain, thus lessening the competitive pressure, and shifting the focus towards quality to enhance the likelihood of being referenced and benefiting—a quality-over-quantity strategy.

Although in-process score feedback did not enhance the quantity or quality of works between the two tasks, within-group comparisons confirm that feedback positively impacted quality, with a stronger stimulus observed among participants using QAM. This may be attributed to increased competitive motivation. The multi-round feedback provided process-oriented guidance, leading participants to identify high-scoring strategies more quickly. Consequently, their design thinking converged faster, resulting in fewer submissions compared to the more divergent approach in Task 1, as they concentrated on producing higher-quality designs. However, this may have also prematurely limited idea divergence, reducing the diversity of creativity.

### Participants' behavior and attitude

The collaborative behavior of "referencing" is a central feature of this experiment. It is a motivation-driven cognitive process through which individuals integrate knowledge by inheriting or improving others' ideas. The most prevalent innovative practices involving references are evident in the realm of academic papers, where its value is well recognized. In design, referencing may reflect recognition of others' knowledge or intrinsic interest in communication. Examining this characteristic facilitates a deeper understanding of how the variables influence designers' motivations, thereby indirectly impacting their referencing behaviors and collaborative outcomes. Drawing on the concept of "building on others' ideas" as "citing inspirations" [5], this paper categorizes referencing into two types based on the source: (1) self-referencing, where participants reference their own previous work; (2) referencing others, where participants reference the work of other members.

**Collaborative network and reference features.** Reference networks were analyzed using UCINET 6 to evaluate creative output and collaboration patterns across experimental groups. We examined network characteristics, reference behavior statistics, and the chain length of scheme references. Nodes represent a designer's total proposals per round (Designer ID_ Round), while each directed, weighted edge points from a referenced proposal to the referencing one (edge weight = reference frequency). Four integrated-round networks were constructed for comparison (Fig 8). The visualizations reveal notable differences in designer behaviors and the flow of ideas. For example, participants under the CA pattern demonstrated significantly higher levels of self-referencing compared to those in the QA pattern, and feedback intervention markedly altered Group 2's collaborative dynamics.

First, we calculate the frequency characteristics of the two types of reference behaviors (Fig 9). In both tasks, Group 2 referenced more frequently than Group 1, suggesting that CAM more effectively promotes interaction and creative exchange among participants. Overall, 90% of the references in Group 1 were made to others' work, while in Group 2, the ratio of self-referencing to referencing others was nearly 1:1, indicating a shift in the primary referencing behavior across tasks. In Task 2, Group 1 experienced a slight decline in total references while still primarily relying on others' work. In contrast, Group 2 exhibited a significant reduction in referencing activity, with nearly all participants decreasing their citations. The balance between self-referencing and referencing others shifted to approximately 7:3, indicating a notable increase in self-referencing, which had previously been overshadowed by external citations.

Table 4 presents network centrality and diffusion metrics. Out-degree (knowledge dissemination) reflects design work referencing by others; in-degree (knowledge absorption) indicates external idea integration. Without feedback, Group 1's in-degree exceeded out-degree by 1.76 times, forming an 'absorption-integration' pattern. Conversely, Group 2 maintained output-oriented collaboration (out-degree = 3.189%, in-degree = 2.892%). When feedback was introduced, both groups

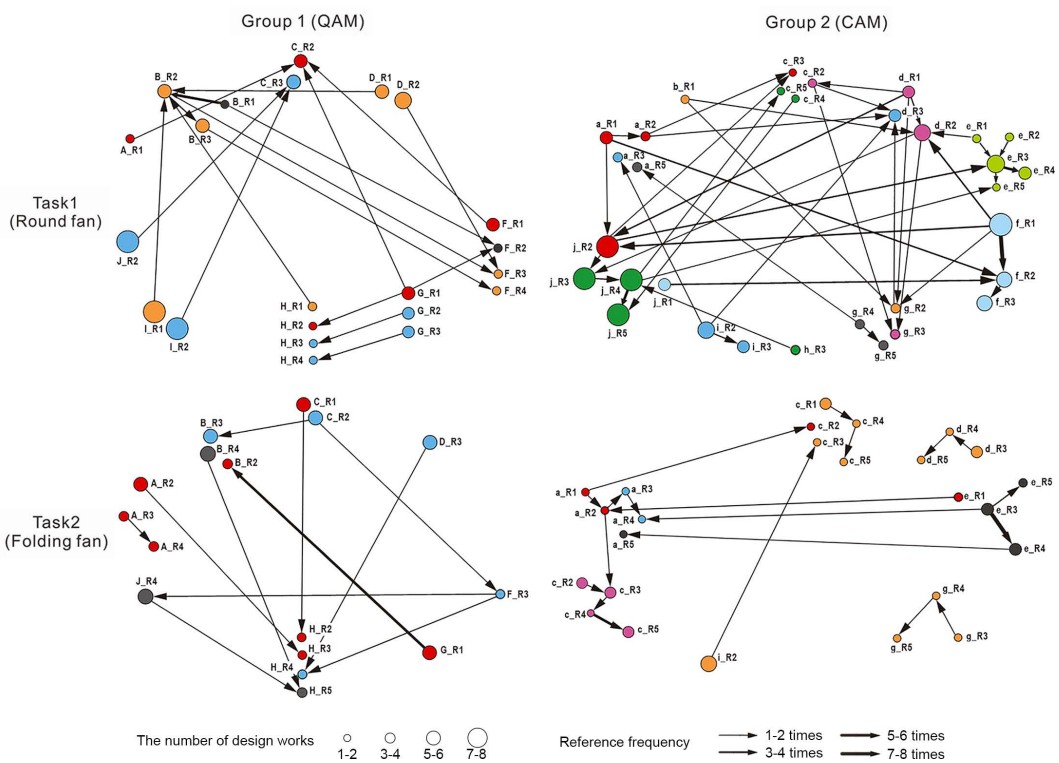

**Fig 8. Reference networks of the four groups.**

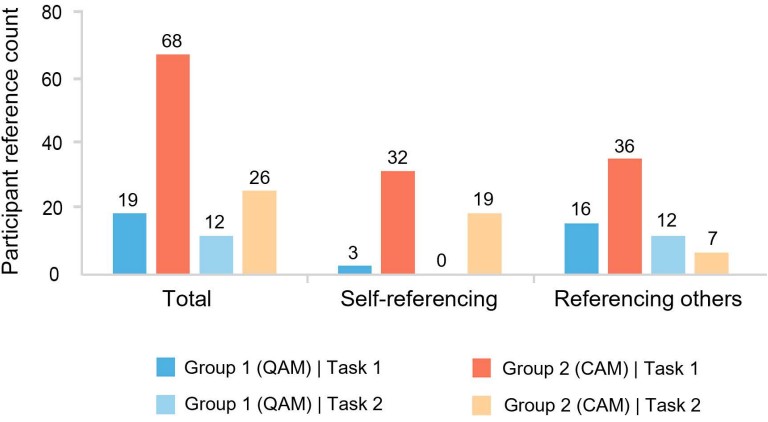

**Fig 9. The reference networks of participants.**

shifted toward output-driven collaboration, with Group 1 developing balanced centrality and bidirectional referencing (both degrees are 1.833%).

Co-hesive subgroup and structural hole analyses uncovered relationships between network position and innovation. Structural hole positions exhibited high centrality, betweenness, and low constraint (e.g., DesignerB_T1R2: Degree = 12.245, Betweenness = 5.782, Constraint = 0.184). In Task 1, subgroups appeared fragmented, and designers

**Table 4. The centrality, cohesive subgroups and structural hole characteristics of the cooperative network.**

| Group | Outdegree | Indegree | Number of Co-hesive Subgroup | Structural Hole |
|---|---|---|---|---|
| Group1 Task1 | 2.728% | 4.810% | 3 | B_R2, C_R2, G_R1 |
| Group1 Task2 | 1.833% | 1.833% | 2 | F_R3 |
| Group2 Task1 | 3.189% | 2.892% | 7 | j_R2, d_R2, e_R3, j_R4, d_R1, d_R3, g_R2 |
| Group2 Task2 | 3.096% | 2.402% | 4 | a_R2 |

Note. The constraint index is less than 0.4 for all structural holes

who served as "bridges" between distinct subgroups were able to integrate diverse design ideas and create unique value. However, this also introduced the risk of dispersed knowledge flow. After feedback intervention, structural holes decreased as designers preferentially engaged with high-scoring proposals within directional idea clusters, consolidating smaller network communities.

Group-level comparisons showed the number of co-hesive subgroups under the QA pattern was approximately half that observed in Group 2. This structural difference may explain why Group 1 were largely limited to single-link reference chains: insufficient diversity in cross-group interactions likely constrained creative evolution.

Regarding structural cohesion, since proposals were allowed to reference those from earlier rounds, the theoretical upper limit of total references could have reached 12,000. However, actual references numbered only hundreds, which made both network density and clustering coefficient approached zero, rendering them statistically insignificant.

Fig 10 illustrates the relationship between the number of references a design received and its score. The average scores indicate a roughly linear positive correlation, suggesting that designs with higher scores tend to achieve higher reference frequencies. In Task 1, no distinct relationship was observed between reference frequency and scores, as participants referenced designs based on personal preference without knowing the scores. In comparison, Task 2 demonstrated a clear positive correlation, with a clear upward trajectory and stronger directional focus. The average score of referenced designs in Task 2 was also higher than in Task 1 (AVG2 = 14.30, AVG1 = 13.87), suggesting that in-process score feedback guided collaborative referencing behavior, reflecting reward-driven motivation. Notably, the highly referenced, high-scoring design e3-1 (referenced four times by designer e himself and once by designer a) appeared in Task 2 under feedback conditions and earned a contribution reward for being referenced by the winning work.

When observing referencing behavior during the experiment, a significant interaction effect was identified in Group 2 during Task 2, which means that the impact of one independent variable on a dependent variable differs depending on the level of another independent variable [85]. As shown in Fig 11, the parallel lines in Group 1 suggest that feedback had a consistent effect on referencing behavior under QAM, with no interaction effect, while the non-parallel lines in Group 2 indicate a notable interaction, particularly affecting the "referencing others" behavior. The comparison shows that self-referencing in Group 2 greatly increased after feedback, revealing a profit-driven tendency not observed in other groups. This may hinder the harmonious development of open-source design collaboration. While CAM enhances design quality and promotes collaboration, the decision to provide in-process score feedback should consider the social context and specific project needs.

The connections between global works and authors throughout the entire collaboration process can also be represented by the length of the "reference chain". Unlike previous studies, this research employs multiple tasks and rounds for horizontal comparison, with a maximum chain length of four links. Fig 12 presents that most reference chains in this experiment were limited to a single link. A few two-link chains appeared in each group, while the only two three-link chains emerged in Group 2 during Task 2. One chain involved both self-referencing and referencing others, linking designers a, e, and j; the other consisted entirely of self-references by designer a (with all designs scoring above 17), and this chain's high-scoring designs also ultimately won awards. A longitudinal analysis of network dynamics across rounds reveals that

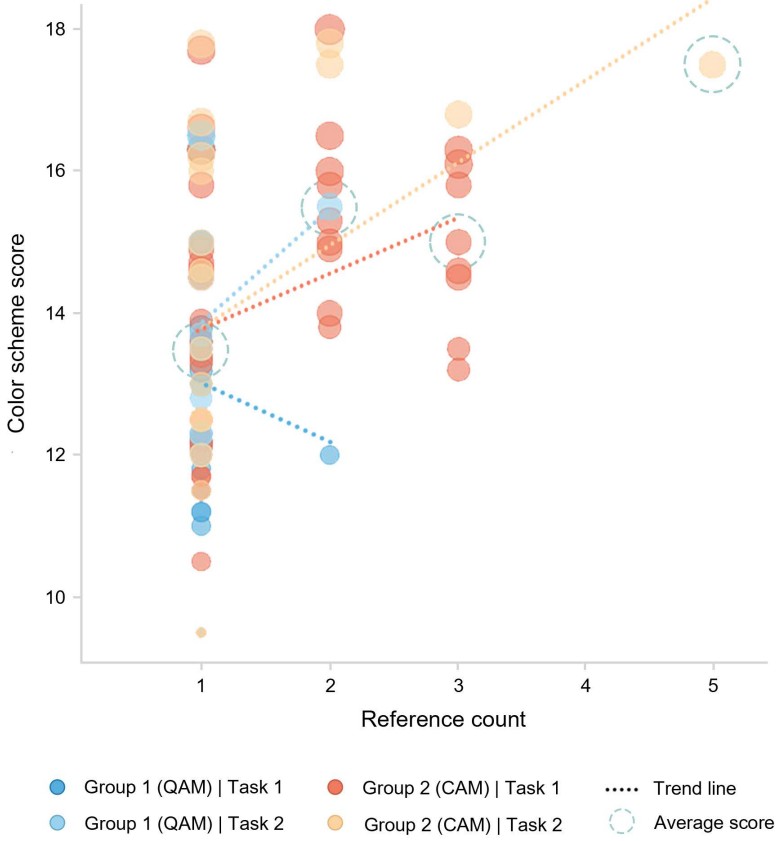

**Fig 10. Relationship between scheme scores and reference counts.**

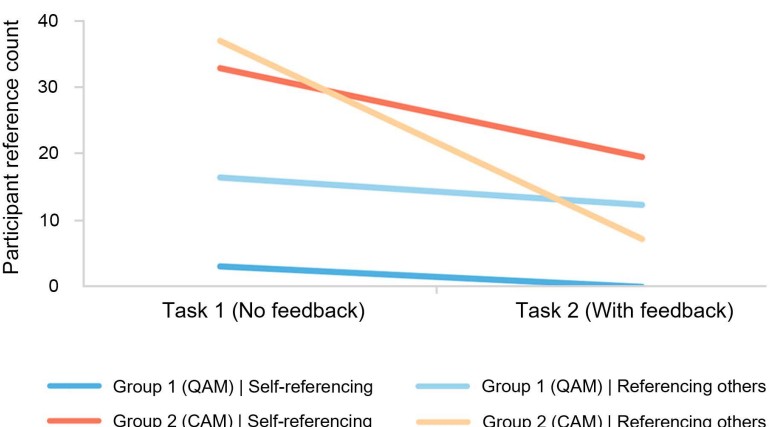

**Fig 11. Interaction effect between in-process feedback and contribution-based allocation mechanism.**

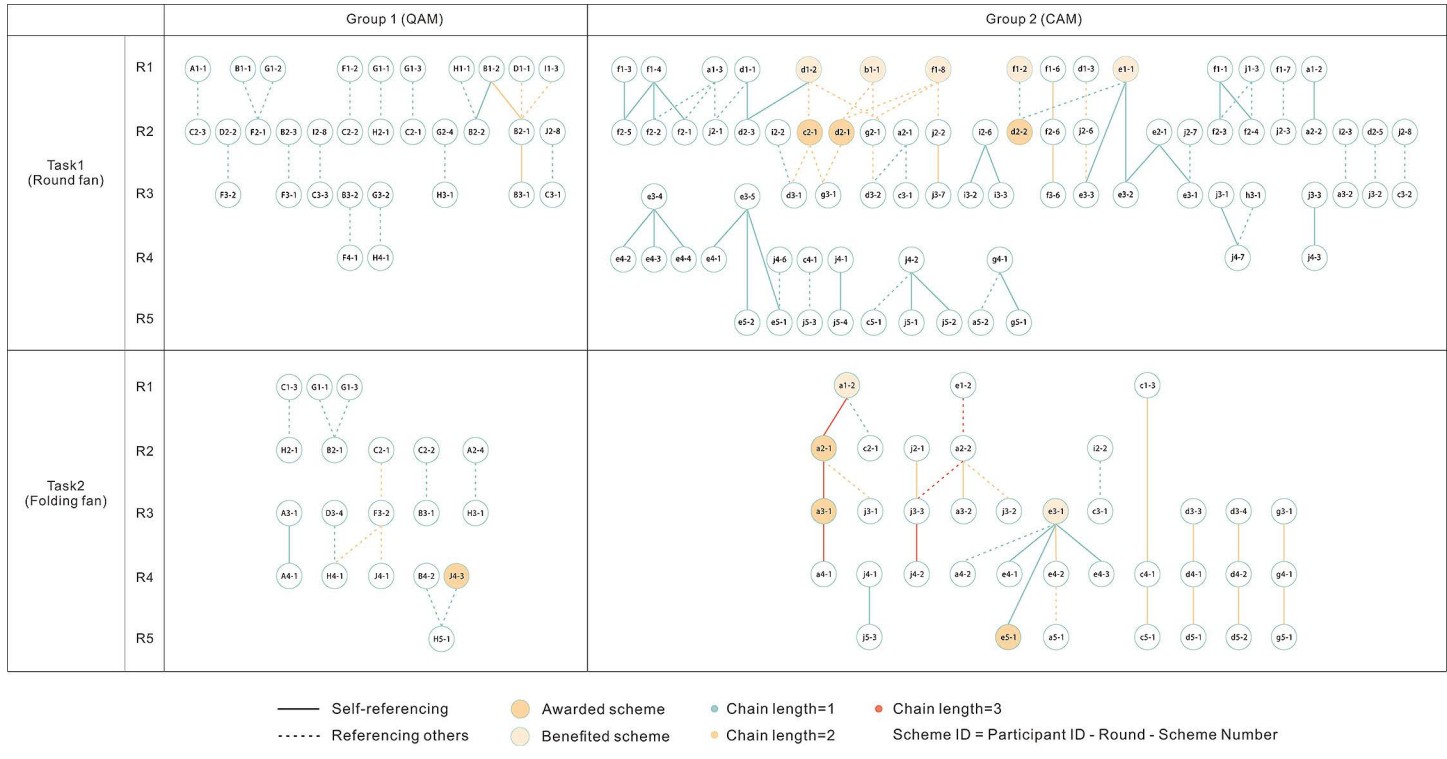

**Fig 12. Reference network of color schemes.**

in the absence of feedback, referencing frequency showed a converging trend (k G1 = −3.67, k G2 = −5.67), and creative diffusion declined over time. This may indicate that designers gradually clarified their direction during collaboration, shifting from external inspiration to internal idea consolidation. In contrast, when feedback was introduced, collaboration frequency reversed from decline to fluctuating growth, offering strong support for the idea that feedback sustains open-source collaboration and fosters the diffusion of creativity.

**Awards of design work.** The winning results are presented in Table 5. A horizontal comparison reveals distinct differences in referencing behavior and outcomes between the two reward mechanisms. The relevant information of the two groups was summarized by individual earnings in descending order (see Table 6).

In Group 1, five participants won awards, with none of the six winning works involved any referencing. Participant G consistently secured the highest rewards despite producing fewer works and engaging in no referencing behavior, suggesting that he may be a professional designer who prioritizes quality over quantity. Contrarily, Participant I had a notably high output (I = 35, AVG = 17), but with average quality (I = 12.22, AVG = 11.99) and no referencing, indicating that he may be a novice designer who focuses on quantity through hard work. Although Participant H frequently referenced others' works, this collaboration did not effectively enhance his own design outcomes, leading to lower output and no awards.

In Group 2, four participants won awards directly, while two others also earned bonuses for their contributions to the reference chains. The winning works frequently exhibited referencing behavior. Participant e, despite producing average output, earned the highest rewards in the group through strategic self-referencing and being referenced by others. Participants a and d each had two winning works; a leveraged extensive self-referencing for additional gains, while d benefited from having his work referenced by the first-place winner, leading to substantial profits. Interestingly, when scores were

**Table 5. Award status of the design works.**

| Task | Group 1 | | | | | Group 2 | | | | |
|---|---|---|---|---|---|---|---|---|---|---|
| | Prize | ID | Reference | Bonus（RMB） | Result | Prize | ID | Reference | Bonus（RMB） | Result |
| Task 1 | 1 | G2-3 | / | 300 | G:300 D:150 I:50 | 1 | c2-1 | d1-2 | c:125 d:125 | d:250 c:125 f:62.5 b:37.5 e:25 |
| | 2 | D2-5 | / | 150 | | 2 | d2-1 | b1-1 f1-8 | d:75 b:37.5 f:37.5 | |
| | 3 | I5-2 | / | 50 | | 3 | d2-2 | f1-2 e1-1 | d:50 f:25 e:25 | |
| Task 2 | 1 | G5-7 | / | 300 | G:300 J:150 F:50 | 1 | e5-1 | e3-1 | 250 | e:250 a:250 |
| | 2 | J4-3 | / | 150 | | 2 | a3-1 | a2-1 a1-2 | 150 | |
| | 3 | F4-1 | / | 50 | | 3 | a2-1 | a1-2 | 100 | |

**Table 6. Participants' award status, design quantity, and reference behavior.**

| Group 1 | | | | | Group 2 | | | | |
|---|---|---|---|---|---|---|---|---|---|
| Designer | Award count | Design quantity | Reference behavior | Bonus（RMB） | Designer | Award count | Design quantity | Reference behavior | Bonus（RMB） |
| **G** | **2** | **M** | **/** | **600** | **e** | **3** | **M** | **SR** | **275** |
| D | 1 | M | / | 150 | a | 4 | M | L | 250 |
| J | 1 | M | L | 150 | d | 3 | M | RO | 250 |
| **I** | **1** | **H** | **/** | **50** | c | 1 | M | RO | 125 |
| F | 1 | M | L | 50 | **f** | **2** | **H&C** | **SR** | **62.5** |
| A | / | M | L | | b | 1 | L | / | 37.5 |
| B | / | M | SR | | g | / | L | L | |
| C | / | M | L | | h | / | L | / | |
| E | / | M | / | | i | / | M | L | |
| **H** | **/** | **L** | **RO** | | **j** | **/** | **H&C** | **H** | |

Note. In terms of quantity, H means high output, M means medium output, L means low output, and C means large variation in output. In reference behavior, L means low references, SR refers to primarily self-referencing, and RO refers to primarily referencing others. Participants in bold proceeded to in-depth interviews.

not disclosed, all three winning works were the result of referencing others, while with score feedback, all the winning works were self-referenced, reducing the number of beneficiaries to two and resulting in a monopoly of rewards.

**Participants' attitudes.** Previous studies on collaboration and reward allocation have extensively discussed the critical role of psychological factors, such as those proposed by self-determination theory. To further explore the reasons and motivations behind collaborative behavior. To further explore the underlying motivations and causes of collaborative behavior, follow-up interviews and questionnaires were conducted after the experiment, which provided insight into the roles of intrinsic motivation, risk preference, and social comparison tendencies. Six individuals were selected for semi-structured interviews: G, H, and I from Group 1, and e, f, and j from Group 2, based on participant characteristics shown in Table 6. The results indicated that all interviewees unanimously agreed that CAM was fairer (QAM = 3.33, CAM = 4.5), but there was disagreement over which allocation method was more attractive, possibly due to individual differences in risk preference. Participant H noted that although QAM had fewer winning slots, the higher amounts made it more competitive

and motivating —indicating a preference for high-risk, high-reward situations, while participant j expressed that CAM resulted in a more evenly distributed reward, increasing the likelihood of winning and making participation more appealing, suggesting a preference for low-risk, stable returns.

The interviewees expressed greater satisfaction with the provision of in-process score feedback. For example, participant I stated, "When my score wasn't high enough, it created a sense of urgency, pushing me to approach the next round more seriously. Additionally, seeing the scores of various works helped guide my design direction to some extent. Otherwise, I would have felt more uncertain, although this uncertainty also kept me alert." It reveals that feedback interventions not only stimulate intrinsic motivation but also activate additional drivers by offering a pathway for social comparison, such as the desire for self-improvement or the anxiety of falling behind. These mechanisms lead individuals to adjust their creative or referencing behaviors, ultimately resulting in differences in both output and patterns of collaboration.

A follow-up questionnaire was developed based on experimental results and interview insights. It was distributed to 20 participants and scored using a five-point Likert scale (1 = strongly disagree to 5 = strongly agree). All collected data were valid. In evaluating the fairness and attractiveness of the reward mechanisms, the two were close, but CAM was considered slightly superior (fairness: QAM = 4.0, CAM = 4.2; attractiveness: QAM = 3.3, CAM = 3.5).

Most participants were satisfied with their own designs, but few were confident about winning. Notably, members of Group 2 had higher average confidence in winning (Group 1 = 2.7, Group 2 = 3.3), indicating that CAM provided some motivational benefits, likely due to its removal of the fixed number of winning slots.

Providing in-process feedback on the scoring reduced participants' willingness to submit more than three designs (Group 1 from 2.4 to 2.0, Group 2 from 2.2 to 1.8), aligning with the overall decrease in submissions observed in Task 2. Questionnaire responses revealed that the main influencing factors were free time, design inspiration, and interest in the task, which suggests that a decline in intrinsic motivation negatively affects the quality and quantity of design outputs. Without in-process score feedback, the number of designs submitted by others was a secondary factor influencing participants' willingness, leading to uncertainty and a herd mentality. With feedback provided, participants focused more on their scores, their risk aversion to uncertainty led to greater goal orientation, reducing the tendency for aimless design.

Additionally, the survey results supported the conclusion that "participants were more satisfied with the provision of in-process score feedback." Overall, the main impacts of score disclosure on participants are summarized in Table 7, probably affect participants by inducing psychological responses through external competitive pressure, self-improvement incentives, and benchmark setting, which in turn influence their creative and collaborative behaviors.

However, attitudes toward profit-seeking and mutual benefit varied. Ideally, everyone would follow the rules to achieve mutual success, but in reality, human nature is unpredictable. Many participants expressed concerns that some might be influenced by profit-seeking motives, leading to unethical behavior and collaboration breakdowns.

Table 7. The impact of providing feedback on participants' satisfaction and its main effects.

| Feedback | Satisfaction | Main effects |
| --- | --- | --- |
| Task1<br>No in-process score feedback | 3.05 | The sense of uncertainty and confusion makes it difficult for novices to judge the quality of their work. |
| | | Selecting reference objects based on personal preference may not align with the judges' requirements. |
| Task2<br>With in-process score feedback | 3.8 | Low scores create pressure and a sense of urgency, motivating better performance in the next round. |
| | | Increases sense of involvement; achieving high scores or seeing score improvement provides a sense of accomplishment and boosts enthusiasm. |
| | | Enhances the desire to win; makes it easier to learn from high-scoring works, encouraging designers to move away from their own style preferences. |
| | | Helps designers better understand and grasp the judges' evaluation criteria and aesthetic preferences. |

**Summary.** In open-source design collaboration, participants commonly involve themselves in interaction. Under the QA reward mechanism, the collaboration network tends to consolidate into an integrated structure, limiting the diversity of communication. In contrast, the CA reward mechanism more effectively promote creative exchange among participants, fostering a knowledge diffusion network that benefits a broader group. This model is viewed as fairer and more attractive. However, the objectivity of collaboration was also influenced by the reward mechanism. CAM tended to induce profit-driven behavior, especially when feedback was introduced, causing the collaboration network to fragment rapidly. To maximize the probability of reward, frequent self-citation emerged within the group, ultimately leading to a high concentration of awards among a few individuals.

The distribution of scores for referenced designs indicates that participants still gravitate towards high-quality work, mitigating the potential loss of diversity from profit-driven behavior. In-process score feedback made evaluators' preferences visible, revealing both participants' own performance and the range of high-scoring designs. Under clearer social comparison and reduced uncertainty, referencing behavior becomes more convergent and purposeful, which facilitates focused collaboration. Participants reported that such feedback enhanced their sense of engagement; intrinsic motivation was activated in each round and influenced subsequent performance, contributing to higher satisfaction levels.

Overall, QAM effectively stimulates participants' competitive desire, but its reliance on a single form of external incentive may lead to "motivation fatigue," limiting its value in sustaining collaboration and creative diffusion. Conversely, the CA pattern excels in fairness, attractiveness, and boosting participants' confidence and willingness to participate. By activating both intrinsic achievement motivation and extrinsic incentives (reward for contribution), this model promotes a "virtuous cycle" that supports interaction and innovation in open-source design collaboration. However, whether to provide in-process score feedback should align with the project's goals.

## Discussion

### Theoretical implications

Previous research has demonstrated that integrating open-source collaboration into design activities is feasible and valuable [45,86]. This study reveals that when applied to product design, a quota allocation without in-process score feedback is more effective in increasing the quantity of design outputs but often at the cost of quality, leading designers to focus more on their own work rather than collaboration. While this approach may boost competition for some, it generally results in lower satisfaction and is less supportive of sustainable project development. In contrast, a contribution-based allocation with feedback enhances design quality and fosters frequent collaboration, leading to higher satisfaction but also potentially introducing profit-driven behavior that could affect collaboration objectivity.

Open-source design collaboration tasks are essentially a form of design competition with transparent processes, where participants still compete independently, but the open access of designs and the allowance for referencing can enhance collaboration [41,42]. As mentioned earlier, a large-scale experiment applied genetic algorithms to guide collaborative sketching for children's chair designs, finding that the number of creative ideas in the final generation was significantly greater than in the first [10]. However, in this study, design output fluctuated across rounds and even declined (as seen in Group 2 during Task 2). This may be attributed to the external conditions introduced in the experiment, which encouraged more unpredictable behaviors, such as strategic gaming, leading to a dual impact of collaboration on output. Similarly, the value of monetary compensation can negatively affect the quantity of contributions [58].

Previous studies have affirmed the positive impact of in-process feedback [25,64]. In this research, the in -process score feedback timely communicated evaluators' preferences, providing guidance that helped participants avoid generating designs blindly. This aligns with Lian Jian et al.'s conclusion that in-process feedback can reduce task uncertainty [66]. They confirmed that whether in the form of numeric ratings or textual comments, such feedback can help attract submissions. However, our study found that in-process feedback did not increase the number of submissions. This difference

may stem from the fact that traditional crowdsourcing typically involves interactions between the project owner and participants, whereas our open-source experiment encouraged interaction among participants. The public disclosure of others' design quantities and scores influenced participants' willingness to produce more work. Additionally, providing in-process feedback caused participants to shift their focus from personal aesthetic preferences to a more utilitarian approach. This led to reduced collaboration, transitioning from broad interaction to a more narrow, focused approach.

Survey responses indicated that participants felt more engaged and motivated with in-process score feedback, reflecting achievement emotions [87]. For example, enjoyment is experienced when a positive evaluation of a task occurs, while anxiety and shame arise in the face of failure or uncertainty [88–90]. In Task 2, visible scores allowed high-scoring designs to guide the team and provide a sense of accomplishment for their creators, while low-scoring designs served as warnings and helped identify mistakes, offering a corrective function. Low-scoring designs brought a sense of defeat to their creators, motivating them to strive harder in the next round. Regardless of the score, feedback provided guidance for the design process and enriched participants' psychological experiences.

Some notable outliers in the results are worth discussing. In Group 2, participants f and j showed a significant reduction in the number of designs (f1 = 28, f2 = 5; j1 = 34, j2 = 13). In follow-up interviews, participant j explained, "With clear goals and a focus on the highest scores, I was able to avoid unnecessary work," indicating a shift from high output to a more focused approach. Participant f provided a more detailed response:

*At first, I made more designs because I found the task fun and had lots of ideas to try out. But later, when I could see my scores, I started focusing on getting a few designs just right and only submitted the ones I really liked. Plus, when I saw others weren't submitting as much, I decided to hold back some of my own work too.*

As social cognitive theory (SCT) proposes that individuals' actions are influenced by an intricate combination of personal convictions, environmental factors, and expected consequences [91]. The participants' responses indicate that they truly understood the importance of score feedback and adjusted their strategies to ensure effort and return rate.

### Practical implications

Several scholars have argued that open source sustainability is a latent key issue [15,92]. This study offers insights for managing open-source design collaboration, communities can benefit in multiple ways. First, project outcomes are influenced by various factors, and rules should be carefully designed based on the core objectives of the crowdsourcing project. For the project initiators, if the goal is to generate a large volume of ideas to inspire creativity, a quota allocation with high rewards can boost competition and increase output, though it may not support long-term sustainability. If the goal is to secure a few high-quality designs for direct use or expand the scope of collaboration as much as possible, a contribution-based allocation encourages collaboration and the development of superior designs, which also promotes continued participation.

Second, for the platform managers, involving evaluators (project owners) in the collaboration process and providing timely feedback is recommended. This facilitates faster convergence of designs, enhances innovation, and enriches designers' experiences, thereby increasing their willingness to participate in future projects. However, caution is needed regarding the simultaneous implementation of the CA pattern and in-process feedback. For open-source design crowdsourcing projects, it is advisable to establish a system integrating reference tags and traceability chains to protect designers' original knowledge and make referencing behaviors an explicit identifier of participants' professional identities. This helps resolve intellectual property disputes and strengthens collaboration networks by linking resource access to contribution.

Finally, for platform designers, project selection should align with personal goals and skills. Those aiming to accumulate work quantity and maximize rewards may prioritize projects with QA pattern, while setting appropriate expectations

for lower returns. Those seeking reputation and creative growth may be more suitable for the CA pattern projects without feedback, which fully integrate collaborative networks and foster frequent user interactions—similar to the GitHub model.

Group communication and interaction can generate collective intelligence [93], further enhancing participants' understanding of tasks and improving the quality of creative output [94]. The experimental results also revealed how differences in participants' motivations affect knowledge contribution, output quality, and sustainability, which are the core issues in crowdsourced innovation. We suggest strengthening the stimulation of intrinsic motivation by autonomy, competence and a sense of relationship in an appropriate way. According to Social Exchange Theory, individuals weigh costs and benefits in social interactions; poorly designed incentives may lead to free-riding or monopolistic behavior, which poses risks to knowledge-sharing economies. We hope to identify effective collaboration models and foster the healthy expansion of collaborative networks. Through such feedback and knowledge sharing, both designers' skills and the overall quality of project community improve. This study offers valuable guidance for both participants and project owners, providing practical guidance for effectively implementing open-source collaboration in product design.

### Limitations and future research directions

As the number of participants in collaborative communities grows, behaviors become more unpredictable, potentially leading to disruptive processes driven by competition [95], including referencing behaviors. Our experiment was conducted under the precondition of participant integrity. However, the lack of large-scale crowd collaboration and the relatively concentrated sample limit the generalizability of our findings to broader crowdsourcing contexts. Results may vary across different professional backgrounds, design tasks (e.g., color matching vs. modeling), or offline settings. We acknowledge the broader utility of large samples in research and recommend attention to the value of small-sample studies for generating reproducible and novel insights [96]. As an exploratory study providing a conceptual framework, this work encourages future research to involve more diverse and larger participant groups to improve data validity and result reliability.

Second, this study focused solely on colorizing tasks, which may not represent all types of designs. Future research could include tasks such as hand-drawing or modeling across various professional domains to develop design management strategies tailored to different task orientations.

Third, the manual tagging of "referencing" behaviors in this study may have disrupted the natural collaborative experience, leading to potential behavioral distortions. More natural methods for tracking these behaviors should be developed to preserve the authenticity of the collaborative environment.

The "score feedback" and "referencing" methods proposed in this study are based on existing collaborative models, but design collaboration is not limited to these approaches. Future research should explore and develop more effective models to meet the needs of different design scenarios.

Finally, individuals with different personalities and abilities respond differently to the same incentives [97–99]. The underlying behavioral motivations are complex and interesting, and further research into the impact of individual differences on open-source collaboration is warranted.

### Conclusion

Within the context of product design, the transition to open and transparent collaboration introduces greater complexity to designers' behaviors, as they become regulated by interests and feedback, ultimately influencing project outcomes and long-term sustainability. Quota allocation promotes high output and motivates participants, while contribution-based allocation fosters closer interaction and higher-quality outcomes through collaborative learning. Although in-process feedback may reduce the number of submissions, it consistently enhances quality, improves participant satisfaction, and provides clearer direction. However, combining in-process feedback with contribution-based allocation can lead to profit-driven behaviors. We believe in the powerful creativity of open-source collaboration and advocate for further exploration of additional influencing factors and barriers. This research provides valuable guidance for developing more

effective collaboration strategies, improving the integration of open-source models into design crowdsourcing platforms, and deepening our understanding of how to leverage collective intelligence for innovative design in practical settings, thereby fostering both individual and community innovation.

## Supporting information

**S1 Data. Expert scoring of the work**
(XLSX)

**S2 Data. Follow-up Questionnaire on Experimental Experience**
(DOCX)

## Acknowledgments

We would like to express our gratitude to the experts for their responsible evaluation of each design throughout the experiment. We also extend our thanks to all researchers for their contributions to the present effort.

## Author contributions

**Conceptualization:** Boqun Xu, Yiting Zhou, Xiaojian Liu.

**Data curation:** Kexin Xu.

**Funding acquisition:** Boqun Xu, Yuning Zhu.

**Investigation:** Boqun Xu, Kexin Xu.

**Methodology:** Boqun Xu.

**Project administration:** Kexin Xu.

**Resources:** Yuning Zhu, Xiaojian Liu.

**Supervision:** Boqun Xu, Yuning Zhu, Xiaojian Liu.

**Validation:** Kexin Xu.

**Visualization:** Kexin Xu, Yiting Zhou.

**Writing – original draft:** Kexin Xu, Yiting Zhou.

**Writing – review & editing:** Boqun Xu, Yuning Zhu, Xiaojian Liu.

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
