## [Decision Letter · Decision Letter 0]

PONE-D-25-04408Driving innovation: When design collaboration becomes open source, how do reward mechanisms and in-process feedback act as catalystsPLOS ONE

Dear Dr. Zhu,

Thank you for submitting your manuscript to PLOS ONE. After careful consideration, we feel that it has merit but does not fully meet PLOS ONE’s publication criteria as it currently stands. Therefore, we invite you to submit a revised version of the manuscript that addresses the points raised during the review process.

We look forward to receiving your revised manuscript.

Kind regards,

Dmitry Kochetkov

Academic Editor

PLOS ONE

**Journal Requirements:**

1. When submitting your revision, we need you to address these additional requirements. Please ensure that your manuscript meets PLOS ONE's style requirements, including those for file naming. The PLOS ONE style templates can be found at https://journals.plos.org/plosone/s/file?id=wjVg/PLOSOne_formatting_sample_main_body.pdf and https://journals.plos.org/plosone/s/file?id=ba62/PLOSOne_formatting_sample_title_authors_affiliations.pdf 2. In the online submission form, you indicated that “Data cannot be shared publicly due to the protection of personal design results. Data are available from the corresponding author on reasonable request.” All PLOS journals now require all data underlying the findings described in their manuscript to be freely available to other researchers, either a. In a public repository, b. Within the manuscript itself, or c. Uploaded as supplementary information.This policy applies to all data except where public deposition would breach compliance with the protocol approved by your research ethics board. If your data cannot be made publicly available for ethical or legal reasons (e.g., public availability would compromise patient privacy), please explain your reasons on resubmission and your exemption request will be escalated for approval. 3. Please include your full ethics statement in the ‘Methods’ section of your manuscript file. In your statement, please include the full name of the IRB or ethics committee who approved or waived your study, as well as whether or not you obtained informed written or verbal consent. If consent was waived for your study, please include this information in your statement as well. 4. Please upload a new copy of Figure 3 as the detail is not clear. Please follow the link for more information: "https://blogs.plos.org/plos/2019/06/looking-good-tips-for-creating-your-plos-figures-graphics/" https://blogs.plos.org/plos/2019/06/looking-good-tips-for-creating-your-plos-figures-graphics/

**Additional Editor Comments:**

This study offers a timely exploration of reward mechanisms and in-process feedback in open design collaboration, employing a structured experimental design to evaluate their impact on creative output and referencing behaviors. While the research addresses a significant gap in understanding collaborative dynamics and presents actionable insights, the reviews highlight critical areas where theoretical, methodological, and analytical refinements are necessary to strengthen the paper’s contribution.

1. The conceptualization of “referencing behavior” as a mediating variable requires deeper theoretical integration. Reviewer 1 rightly notes that the current classification (self- vs. other-reference) remains descriptive. Anchoring this construct in established motivational frameworks (e.g., attribution theory, social identity theory) would enhance explanatory power and generalizability.

2. Similarly, the causal pathways connecting motivation, behavior, and outcomes need elaboration. Integrating questionnaire data to model psychological mechanisms (e.g., intrinsic motivation, social expectations) could bridge this gap.

3. The small, context-specific sample (N=20) limits the external validity of findings. While the experimental design controls for internal validity, the paper should explicitly address how institutional or demographic homogeneity might constrain broader applicability. Expanding the “Limitations” section to discuss these factors is essential.

4. As noted by Reviewer 2, providing descriptive statistics on participant characteristics (e.g., prior collaboration experience, domain expertise) and testing for baseline differences between experimental groups would bolster confidence in the results’ integrity.

5. The current focus on citation path length overlooks richer network dynamics (e.g., centrality, cohesion). Incorporating metrics from social network analysis could better explain how reference structures mediate creative collaboration.

6. The study could leverage questionnaire responses more systematically to unpack psychological drivers behind observed behaviors, thereby enriching the causal narrative.

7. The “Discussion” section should articulate clearer managerial implications. For instance, how might platform designers balance reward structures with feedback mechanisms to optimize creativity? How do referencing behaviors inform policies for knowledge attribution in open collaborations? Addressing these questions would enhance relevance for applied audiences.

8. The paper’s academic contribution could be sharpened by situating findings within broader debates on motivation in crowdsourced innovation or knowledge-sharing economies.

Reviewers' comments:

Reviewer's Responses to Questions

**Comments to the Author**

1. Is the manuscript technically sound, and do the data support the conclusions?

Reviewer #1: Yes

Reviewer #2: Yes

2. Has the statistical analysis been performed appropriately and rigorously? 

Reviewer #1: Yes

Reviewer #2: Yes

3. Have the authors made all data underlying the findings in their manuscript fully available?

Reviewer #1: Yes

Reviewer #2: No

4. Is the manuscript presented in an intelligible fashion and written in standard English?

Reviewer #1: Yes

Reviewer #2: Yes

5. Review Comments to the Author

**Reviewer #1: ** This study focuses on two critical variables in open design collaboration: reward mechanisms and in-process feedback. Through a 2×2 experimental design within a structured creative task, the paper systematically examines their influence on output quantity, quality, and referencing behaviors. The research demonstrates theoretical relevance and practical implications. However, several aspects of the theoretical framing and empirical execution warrant further refinement.

1. The theoretical construction of key variables appears somewhat empirical.

“Referencing behavior,” as a central mediating variable, is primarily categorized based on observational distinctions (self-reference vs. other-reference), without being grounded in a broader or transferable motivational framework. The current treatment leans toward descriptive classification rather than explanatory modeling. Stronger alignment with motivation theories—such as attribution theory or social identity theory—would enhance the conceptual robustness.

2. Limitations in sample size and generalizability.

The study involves a relatively small sample (N=20) drawn from a single institutional or organizational context. This raises concerns about the generalizability of the findings across broader populations, platforms, or collaborative scenarios. This limitation should be more explicitly addressed in the “Limitations” section.

3. Causal mechanisms linking motivation, behavior, and outcomes are underdeveloped.

While the experimental results capture the behavioral effects of reward and feedback, the underlying psychological mechanisms—such as risk aversion, social expectations, or intrinsic drive—remain insufficiently explored. A more systematic integration of the questionnaire results could help model these mediating pathways more rigorously.

4. The analysis of reference network structure lacks methodological depth.

The current description of the “reference chain” primarily focuses on the length of citation paths, without incorporating network-analytic metrics such as node centrality, diffusion dynamics, or structural cohesion. This limits the explanatory power of the reference network in mediating creative output and collaborative engagement.

5. The discussion and managerial implications could be further developed.

While the paper presents interesting findings, the practical takeaways for platform designers, innovation managers, or policy makers are somewhat limited. Strengthening this section would enhance the study's relevance to both academic and applied audiences.

**Reviewer #2: ** First of all this post is very interesting and I have some questions about the research methodology and data presentation. First of all, can you explain the strengths of the methodology of this study, why it is suitable for this study and can you clearly answer the research questions of the article. The second is, can the paper show the descriptive characteristics of the experimental subjects, or the differences between the two groups, especially the characteristics that may affect the results of the experiment, thus increasing the credibility of the results.

6. PLOS authors have the option to publish the peer review history of their article (what does this mean? ). If published, this will include your full peer review and any attached files.

**Do you want your identity to be public for this peer review?** For information about this choice, including consent withdrawal, please see our Privacy Policy .

Reviewer #1: No

Reviewer #2: No

---

## [Author Response · Author response to Decision Letter 1]

9 Jun 2025

Dear Editors,

We thank all the editors and reviewers for the valuable comments and suggestions. We have carefully revised the manuscript to enhance its clarity and facilitate the understanding of the readers. Our point-to-point responses are presented in the file named Response to Reviewers. We hope that the revision would satisfactorily address the comments and concerns of the editors and reviewers.

Response to Editor

Summary Comment 1

The conceptualization of “referencing behavior” as a mediating variable requires deeper theoretical integration. Reviewer 1 rightly notes that the current classification (self- vs. other-reference) remains descriptive. Anchoring this construct in established motivational frameworks (e.g., attribution theory, social identity theory) would enhance explanatory power and generalizability.

Response S1: Thank you for your summary.

We appreciate that this thoughtful concern aligns with the query raised by Reviewer 1. The corresponding revisions have been thoroughly addressed in Response 1.1.

Summary Comment 2

Similarly, the causal pathways connecting motivation, behavior, and outcomes need elaboration. Integrating questionnaire data to model psychological mechanisms (e.g., intrinsic motivation, social expectations) could bridge this gap.

Response S2: Thank you for the comment.

We recognize that psychological mechanisms do play an important role in this study. According to this question, we have systematically examined mechanisms related to rewards and competition, and supplemented theories such as intrinsic motivation, risk preference, and social comparison tendencies. For instance, the interviewees' scores of fairness and attractiveness might reflect an individual's risk preference. The interviewees' attitudes towards feedback intervention indicated the adjustment of their internal driving forces to their personal creation or citation behaviors, ultimately leading to differences in the outcome of the work and the collaborative relationship. The specific answer can be seen in Response 1.3.

Summary Comment 3

The small, context-specific sample (N=20) limits the external validity of findings. While the experimental design controls for internal validity, the paper should explicitly address how institutional or demographic homogeneity might constrain broader applicability. Expanding the “Limitations” section to discuss these factors is essential.

Response S3: Thank you for your pertinent advice.

We sincerely appreciate your careful attention to these aspects, which directly reflect comment 1.2. We have explicitly outlined the potential limitations of our research findings and corresponding precautions (see lines 757-765 on page 34), and the specific modifications are as follows in the blue part:

As the number of participants in collaborative communities grows, behaviors become more unpredictable, potentially leading to disruptive processes driven by competition [95], including referencing behaviors. Our experiment was conducted under the precondition of participant integrity. However, the lack of large-scale crowd collaboration and the relatively concentrated sample limit the generalizability of our findings to broader crowdsourcing contexts. Results may vary across different professional backgrounds, design tasks (e.g., color matching vs. modeling), or offline settings. We acknowledge the broader utility of large samples in research and recommend attention to the value of small-sample studies for generating reproducible and novel insights [1]. As an exploratory study providing a conceptual framework, this work encourages future research to involve more diverse and larger participant groups to improve data validity and result reliability.

[1] Li X, Liu J, Gao W, Cohen GL. Challenging the N-Heuristic: Effect size, not sample size, predicts the replicability of psychological science. PLoS ONE. 2024 Aug 23;19(8): e0306911.

Summary Comment 4

As noted by Reviewer 2, providing descriptive statistics on participant characteristics (e.g., prior collaboration experience, domain expertise) and testing for baseline differences between experimental groups would bolster confidence in the results’ integrity.

Response S4: Thank you for the comment.

Based on the suggestion of Reviewer 2, the descriptive data of the experimental personnel have been supplemented accordingly (see Table 1). Potential variables have been controlled as much as possible during the recruitment of subjects and the grouping of personnel to minimize the baseline differences to the greatest extent.

Summary Comment 5

The current focus on citation path length overlooks richer network dynamics (e.g., centrality, cohesion). Incorporating metrics from social network analysis could better explain how reference structures mediate creative collaboration.

Response S5: Thank you for your reminder.

We believe this is consistent with the question raised by Reviewer 1. The specific answer can be seen in Response 1.4. Overall, we have conducted additional data analysis, and the new results are now incorporated in the revised manuscript to provide further support and enhance the robustness of our findings.

Summary Comment 6

The study could leverage questionnaire responses more systematically to unpack psychological drivers behind observed behaviors, thereby enriching the causal narrative.

Response S6: Thank you for the comment.

As the questions raised in summary comment 2, we carefully examined the relationship between the questionnaire and psychological driving factors in the revised draft, the answer and specific modifications here can be referred to Response 1.3.

Summary Comment 7

The “Discussion” section should articulate clearer managerial implications. For instance, how might platform designers balance reward structures with feedback mechanisms to optimize creativity? How do referencing behaviors inform policies for knowledge attribution in open collaborations? Addressing these questions would enhance relevance for applied audiences.

Response S7: Thank you for your specific explanation.

We review and find that the original practical significance indeed did not provide sufficient suggestions for the application audience. We summarize the research conclusions and provide referenceable action suggestions for project initiators, platform managers and designers respectively, to help them improve efficiency or achieve better results in actual implementation as much as possible. The modified contents are as follows:

Several scholars have argued that open source sustainability is a latent key issue [15,92]. This study offers insights for managing open-source design collaboration, communities can benefit in multiple ways. First, project outcomes are influenced by various factors, and rules should be carefully designed based on the core objectives of the crowdsourcing project. For the project initiators, if the goal is to generate a large volume of ideas to inspire creativity, a quota allocation with high rewards can boost competition and increase output, though it may not support long-term sustainability. If the goal is to secure a few high-quality designs for direct use or expand the scope of collaboration as much as possible, a contribution-based allocation encourages collaboration and the development of superior designs, which also promotes continued participation.

Second, for the platform managers, involving evaluators (project owners) in the collaboration process and providing timely feedback is recommended. This facilitates faster convergence of designs, enhances innovation, and enriches designers' experiences, thereby increasing their willingness to participate in future projects. However, caution is needed regarding the simultaneous implementation of the CA pattern and in-process feedback. For open-source design crowdsourcing projects, it is advisable to establish a system integrating reference tags and traceability chains to protect designers' original knowledge and make referencing behaviors an explicit identifier of participants' professional identities. This helps resolve intellectual property disputes and strengthens collaboration networks by linking resource access to contribution.

Finally, for platform designers, project selection should align with personal goals and skills. Those aiming to accumulate work quantity and maximize rewards may prioritize projects with QA pattern, while setting appropriate expectations for lower returns. Those seeking reputation and creative growth may be more suitable for the CA pattern projects without feedback, which fully integrate collaborative networks and foster frequent user interactions—similar to the GitHub model. (page 32, lines 714-737)

Summary Comment 8

The paper’s academic contribution could be sharpened by situating findings within broader debates on motivation in crowdsourced innovation or knowledge-sharing economies.

Response S8: Thank you for your valuable suggestions on the research contribution.

Your question is of summary significance, guiding us to summarize the relevant analyses of the psychological mechanisms and motivation theories in this revision. The detailed analysis can be found in , and we have also included a summary discussion in the Practical Implications section, the details are as follows. We sincerely thank you again for your effort.

Group communication and interaction can generate collective intelligence [93], further enhancing participants' understanding of tasks and improving the quality of creative output [94]. The experimental results also revealed how differences in participants’ motivations affect knowledge contribution, output quality, and sustainability, which are the core issues in crowdsourced innovation. We suggest strengthening the stimulation of intrinsic motivation by autonomy, competence and a sense of relationship in an appropriate way. According to Social Exchange Theory, individuals weigh costs and benefits in social interactions; poorly designed incentives may lead to free-riding or monopolistic behavior, which poses risks to knowledge-sharing economies. We hope to identify effective collaboration models and foster the healthy expansion of collaborative networks. Through such feedback and knowledge sharing, both designers' skills and the overall quality of project community improve. This study offers valuable guidance for both participants and project owners, providing practical guidance for effectively implementing open-source collaboration in product design. (page 33, lines 740-748)

Response to Reviewer #1

Comment 1.1

The theoretical construction of key variables appears somewhat empirical.

"Referencing behavior," as a central mediating variable, is primarily categorized based on observational distinctions (self-reference vs. other-reference), without being grounded in a broader or transferable motivational framework, The current treatment leans toward descriptive classification rather than explanatory modeling, Stronger alignment with motivation theories-such as attribution theory or social identity theory-would enhance the conceptual robustness.

Response 1.1: Thank you for your helpful suggestions.

The original exposition was too simple and subjective. Reference is indeed related to attribution and personal identification. In the revised version, we described in detail the introduction basis and significance of the concept of "reference behavior" at the beginning of Participants' behavior and attitude. For details, please refer to lines 395-403 on page 18.

Comment 1.2

Limitations in sample size and generalizability.

The study involves a relatively small sample (N=20) drawn from a single institutional or organizational context. This raises concerns about the generalizability of the findings across broader populations, platforms, or collaborative scenarios. This limitation should be more explicitly addressed in the limitations section.

Response 1.2: Thank you for the pertinent comment.

We originally briefly mentioned the limitations but not in detail. Due to the limitation of the sample size, it is impossible to calculate the effect size to improve the reliability. In the revised manuscript, we clearly stated the possible directions of limitation and precautions of the research results (Page 34. lines 757-765).

Comment 1.3

Causal mechanisms linking motivation, behavior, and outcomes are underdeveloped.

While the experimental results capture the behavioral effects of reward and feedback, the underlying psychological mechanisms-such as risk aversion, social expectations, or intrinsic drive-remain insufficiently explored, A more systematic integration of the questionnaire results could help model these mediating pathways more rigorously.

Response 1.3: Thank you for your question and recommendation.

We acknowledge the important role of psychological factors in shaping participants' behaviors and outcomes. Following your suggestions, we have systematically investigated the psychological mechanisms related to rewards and competition, and supplemented theories such as intrinsic motivation, risk preference, and social comparison tendencies. We optimized the conclusions based on the original questionnaire data and further linked the potential role of psychological mechanisms on behavior and outcomes. (All the yellow sections in the revised manuscript, pages 25-27, lines 554-606).

Initially, we did not emphasize this aspect due to the extensive existing research on psychological factors in reward and feedback. Our primary focus was on the role of variables in behavior and results, and relatively weakens the mediating role of statistical methods in observing psychological mechanisms. However, as originally mentioned in the overview section of Reward Mechanisms (page 6), intrinsic motivation and external rewards can affect outcome quality (e.g., achievement, fairness expectations). And in-process feedback may stimulate competitive psychology and bring about positive or negative effects (page 8). Still, questionnaires and interviews were conducted and interviews were conducted during the experiment. We believe that such data are equally significant for the research.

Comment 1.4

The analysis of reference network structure lacks methodological depth.

The current description of the "reference chain" primarily focuses on the length of citation paths without incorporating network-analytic metrics such as node centrality, diffusion dynamics, or structural cohesion. This limits the explanatory power of the reference network in mediating creative output and collaborative engagement.

Response 1.4: Thank you for your valuable suggestions.

We think this is an excellent suggestion. We have conducted additional data analysis, and the new results are now incorporated in the revised manuscript to provide further support and enhance the robustness of our findings. To make it clear, we have added the above data and discussions in the main text (All the yellow sections in the revised manuscript from pages 18-23, lines 406 to 514).

In addition, after adding new analysis, we adjusted the sequence of the paragraph structure to optimize the logic. We organized the new network analysis and the original citation chain length analysis into Collaborative network and reference features (page 18), and separately adjusted the original award result analysis to Awards of design work (page 23).

We have also made certain supplements in the summary section (All the yellow sections in the revised manuscript on pages 28-29, lines 617 to 642). Hopefully, you would be satisfied with these supplements.

Comment 1.5

The discussion and managerial implications could be further developed.

While the paper presents interesting findings, the practical takeaways for platform designers, innovation managers, or policy makers are somewhat limited. Strengthening this section would enhance the study's relevance to both academic and applied audiences.

Response 1.5: Thank you for your helpful suggestions.

We review and find that the original practical significance indeed did not provide sufficient suggestions for the application audience. We summarize the research conclusions and provide referenceable

---

## [Decision Letter · Decision Letter 1]

Driving innovation: When design collaboration becomes open source, how do reward mechanisms and in-process feedback act as catalysts

PONE-D-25-04408R1

Dear Dr. Zhu,

We’re pleased to inform you that your manuscript has been judged scientifically suitable for publication and will be formally accepted for publication once it meets all outstanding technical requirements.

Kind regards,

Dmitry Kochetkov

Academic Editor

PLOS ONE

Additional Editor Comments (optional):

The reviewers and I appreciate the comprehensive and thoughtful revisions you have made in response to the previous feedback. Your efforts have significantly strengthened the manuscript and addressed the core concerns raised during the review process.

However, one minor point was noted during the final editorial check:

Peer Reviewer Comment:

"The author has made more comprehensive revisions based on the previous feedback. A small suggestion: The low clarity of the images affects reading, and the explanation of the illustrations needs to be further strengthened, especially the node meanings in the network diagrams of Figure 8 and Figure 12."

Could you please address this comment before final submission?

Reviewers' comments:

Reviewer's Responses to Questions

**Comments to the Author**

1. If the authors have adequately addressed your comments raised in a previous round of review and you feel that this manuscript is now acceptable for publication, you may indicate that here to bypass the “Comments to the Author” section, enter your conflict of interest statement in the “Confidential to Editor” section, and submit your "Accept" recommendation.

Reviewer #1: All comments have been addressed

Reviewer #2: All comments have been addressed

2. Is the manuscript technically sound, and do the data support the conclusions?

Reviewer #1: Yes

Reviewer #2: Yes

3. Has the statistical analysis been performed appropriately and rigorously? 

Reviewer #1: Yes

Reviewer #2: Yes

4. Have the authors made all data underlying the findings in their manuscript fully available?

Reviewer #1: Yes

Reviewer #2: Yes

5. Is the manuscript presented in an intelligible fashion and written in standard English?

Reviewer #1: Yes

Reviewer #2: Yes

6. Review Comments to the Author

Reviewer #1: The author has made more comprehensive revisions based on the previous feedback. A small suggestion: The low clarity of the images affects reading, and the explanation of the illustrations needs to be further strengthened, especially the node meanings in the network diagrams of Figure 8 and Figure 12.

Reviewer #2: (No Response)

7. PLOS authors have the option to publish the peer review history of their article (what does this mean? ). If published, this will include your full peer review and any attached files.

**Do you want your identity to be public for this peer review?** For information about this choice, including consent withdrawal, please see our Privacy Policy .

Reviewer #1: No

Reviewer #2: No

---

## [Editor Report · Acceptance letter]

PONE-D-25-04408R1

PLOS ONE

Dear Dr. Zhu,

I'm pleased to inform you that your manuscript has been deemed suitable for publication in PLOS ONE. Congratulations! Your manuscript is now being handed over to our production team.

Kind regards,

on behalf of

Dr. Dmitry Kochetkov

Academic Editor

PLOS ONE